# A Lac Repressor-Inducible Baculovirus Expression Vector for Controlling Adeno-Associated Virus Capsid Ratios

**DOI:** 10.3390/v16010051

**Published:** 2023-12-28

**Authors:** Jeffrey Slack, Christopher Nguyen, Amanda Ibe-Enwo

**Affiliations:** 1Voyager Therapeutics, 64 Sidney St., Cambridge, MA 02139, USA; audumma@vygr.com; 2Stylus Medicine, Inc., 200 Berkeley St., Boston, MA 02116, USA; chris.nguyen@stylusmedicine.com

**Keywords:** baculovirus expression vector, inducible expression, rAAV, lac repressor

## Abstract

The baculovirus expression vector (BEV) system is an efficient, cost-effective, and scalable method to produce recombinant adeno-associated virus (rAAV) gene therapy vectors. Most BEV designs emulate the wild-type AAV transcriptome and translate the AAV capsid proteins, VP1, VP2, and VP3, from a single mRNA transcript with three overlapping open reading frames (ORFs). Non-canonical translation initiation codons for VP1 and VP2 reduce their abundances relative to VP3. Changing capsid ratios to improve rAAV vector efficacy requires a theoretical modification of the translational context. We have developed a Lac repressor-inducible system to empirically regulate the expression of VP1 and VP2 proteins relative to VP3 in the context of the BEV. We demonstrate the use of this system to tune the abundance, titer, and potency of a neurospecific rAAV9 serotype derivative. VP1:VP2:VP3 ratios of 1:1:8 gave optimal potency for this rAAV. It was discovered that the ratios of capsid proteins expressed were different than the ratios that ultimately were in purified capsids. Overexpressed VP1 did not become incorporated into capsids, while overexpressed VP2 did. Overabundance of VP2 correlated with reduced rAAV titers. This work demonstrates a novel technology for controlling the production of rAAV in the BEV system and shows a new perspective on the biology of rAAV capsid assembly.

## 1. Introduction

Adeno-Associated Viruses (AAVs) are Parvoviruses belonging to the Dependovirus genus and require co-infection of another DNA virus type such as an adenovirus, herpesvirus, or papillomavirus [1,2,3]. AAVs have single-stranded DNA genomes that exist as episomes in the nucleus or as proviruses in chromosome 19 of humans [4]. There are 13 characterized primate AAV serotypes with different tissue tropisms and potencies [5]. AAV capsids are icosahedral with T = 1 symmetry and 60 viral proteins (VPs) per capsid. These 60 VPs are a mixture of VP1, VP2, and VP3 proteins that share common amino acid sequences. The smallest VP3 protein dominates in abundance in assembled capsids. In recent years, rAAVs have been harnessed as gene therapy vectors, with several therapeutics being authorized for clinical use in the United States (Roctavin^TM^, Glybera^TM^, Luxturna^TM^, and Zolgensma^TM^). There have been more than 100 clinical trials of different rAAV drug products [6], and genetically modified capsid serotypes with improved tissue specificities and potencies are on the horizon for a new generation of rAAV therapeutics [7,8].

rAAV gene therapeutics involve expressing AAV replicase (Rep) and capsid (Cap) proteins in the presence of a therapeutic transgene flanked by AAV inverted terminal repeat sequences (ITRs). The result is a rAAV capsid with a packaged transgene. The rAAV is inoculated into the patient, where it transduces targeted cell types. Transgenes are delivered into the cell nucleus, where they are converted into stable double-stranded DNA episomes from which the transgene-encoded therapeutic gene product is expressed.

Application of the insect cell line-based BEV system to produce rAAV therapeutics was first described as a triple BEV infection by [9] and later as a dual BEV infection [10]. The BEV system is advantageous for scaled, high transgene titer production of rAAV therapeutics with unlimited cell culture production capacity, economically amplified BEV inoculums, and the absence of endotoxins or potential human pathogens. The dual BEV infection method to produce rAAV in the BEV system involves the co-infection of a RepCap BEV with an ITR transgene BEV. The RepCap-expressing BEV is engineered with Rep and Cap genes pointing in opposite directions and expressed from very late baculovirus promoters (Figure 1). Rep78 and Rep52 proteins are translated from a common ORF, with Rep52 having its own internal translational initiation codon. The Cap gene encodes for VP1 (81 kDa), VP2 (67 kDa), and VP3 (61 kDa) Cap proteins on a common ORF (Figure 1). The Cap gene also encodes an out-of-frame ORF for the 21 kDa assembly-activating protein (AAP). AAP is essential for capsid assembly in multiple AAV serotypes, including AAV9 [11,12]. In the BEV system, VP1, VP2, and AAP protein ORFs have non-canonical, non-ATG translational start codons. The abundances of translated VP1, VP2, and AAP proteins are less than those of VP3, which has a canonical ATG translational start codon.

A challenge of producing potent rAAV capsids using the BEV system has been optimizing VP capsid ratios. The ideal VP1:VP2:VP3 ratios for AAV capsids are imprecise and have been reported as 1:1:20 [13], 1:1:10 [9,14], and 1:1:8 [15,16]. The method of analysis used to determine VP ratios often differs, and there may be variations in ideal VP ratios for different AAV serotypes. Furthermore, there are novel tissue-specific rAAV capsid variants [8], which may have different optimal VP ratios. Varying the ratio of VPs has been reported to affect the titer and potency of assembled rAAV capsids [17]. Methods for changing VP ratios produced using BEVs have often involved the modulation of the translational context of VP1 [17,18].

The present work describes a method for expressing rAAV capsid VPs at different ratios for a neurospecific AAV9 capsid variant in the BEV system. Instead of expressing VP1, VP2, and VP3 from a common ORF, we separated VP1, VP2, and VP3 ORFs to be expressed independently in a single BEV construct. The VP1 and VP2 ORFs were placed under *E. coli* lac repressor (LacR)-induced regulation in the context of the BEV expression system. VP1 and VP2 genes were designed to separately have Lac Operator (LacO)-containing baculovirus promoters. We show how controlling the abundances of VP1 and VP2 relative to VP3 affects the potency, yield, and transgene titer of rAAV capsids. Overabundance of VP1 and VP2 negatively impacts capsid’s assembly and reduces potency.

## 2. Materials and Methods

### 2.1. Insect Cells

*Spodoptera frugiperda* (*Sf9*) cells used were a Voyager Therapeutics clonal isolate of the *Sf9* cell line [19], which was a clonal derivative of the IPLB-SF-21-AE cell line [20]. *Sf9* were cultured at 28 °C in ESF AF™ Insect Cell Culture Medium (#99-300-01, Expression Systems, Davis, CA, USA). *Sf9* cells were grown at 25 mL scale in Mini Bioreactor tubes (#431720, Corning, Tewksbury, MA, USA) and shaken at 250 rpm or grown in 2 L shake flasks (#431255, Corning) and shaken at 95 rpm. Human embryonic kidney cells were from the HEK 293T cell line (ATCC CRL-1573). These cells were maintained as adherent cultures in Dulbecco’s Modified Eagle Medium (DMEM) (#10564011, Thermo Fisher Scientific, Waltham, MA, USA).

### 2.2. Virus Stocks

Bacmid DNA was generated in NEB 10-Beta *E. coli* (#C3020K, New England Biolabs, Ipswich, MA, USA) and purified by the Qiagen Large Construct Kit (#12462, Qiagen, Germantown, MD, USA) and transfected into *Sf9* cells to be converted into replicating baculoviruses, which we still refer to as bacmids. Bacmid DNA-transfection complexes were prepared by combining 200 ng of bacmid DNA, 5 µL of TransIT-Insect Transfection Reagent (# MIR 6100, Mirus Bio, Madison, WI, USA), and 200 µL of Grace’s Salts (#G8142, Sigma-Aldrich, St. Louis, MO, USA) and transfecting 30 mL of *Sf9* cells at 2 × 10^6^ cells/mL. At 5 days post-transfection, bacmid BEV inoculum stocks were made as baculovirus-infected insect cells (BIICs). The infected cells were pelleted by centrifugation (5 min, 300× *g*) and resuspended at 2 × 10^7^ cells/mL in the original budded virus media. An equal volume of 10% *v*/*v* dimethyl sulfoxide and 5% *w*/*v* trehalose in ESF AF^TM^ media was added, and the resulting BIIC aliquots were stored at −80 °C. BIIC infectious titers were determined using methods described by Nguyen et al., 2023 [21]. These titers were used to determine the multiplicity of infection (MOI).

### 2.3. Constructs

#### 2.3.1. Rep and Cap and LacR Gene Sources

The Rep gene used was based on the AAV2 serotype (NCBI:txid10804). The Cap gene used in this study was a synthetically engineered neurotropic capsid variant with 98% identity to the AAV9 serotype (NCBI:txid235455). It was used as a template for Cap ORFs VP1, VP2, and VP3. The LacR ORF used was based on GenBank J01636.1 and was engineered to encode an N-terminal SV40 nuclear localization signal as described by Slack et al., 1997 [22].

#### 2.3.2. Parental Bacmid VoyBac1.1

All recombinant bacmids in this study were based on the Voyager Therapeutics bacmid called VoyBac1.1 (Figure 2B). The acronym “VB” refers to VoyBac1.1 and is used in front of bacmid construct names. VoyBac1.1 is a “bacmid” type BEV that is based on a bacterial artificial chromosome like the first bacmid called bMON14272 [23] (Figure 2A). Bacmids can be modified and amplified in *E. coli*. When the bacmid DNA is transfected into insect cells, such as *Sf9*, it replicates as a baculovirus. The bacmid bacterial gene elements are retained by the baculoviruses that are generated in insect cells, and there is even the option to transform baculovirus DNA back into *E. coli*, where it will replicate once again as a bacmid. Like bMON14272, the VoyBac1.1 bacmid genome contains a Kanamycin resistance gene, a Tn7 transposition site, and a bacterial mini-F replicon in the genome for the baculovirus *Ac*MNPV E2 (GenBank KM667940.1). Tn7 transposition and bacterial gene elements are located between the *Ac*MNPV genes called ORF603 and ORF1629. VoyBac1.1 differed from the bMON14272 bacmid by being modified to lack the *v-cath* gene [24]. Deletion of the *v-cath* gene and the adjacent *ChiA* gene is described in Nguyen et al., 2023 [21] and is illustrated in Figure 1B. Bacmids that have the *v-cath* gene are poor for rAAV production as the V-CATH protease degrades several AAV capsid serotypes [25], including the neurotropic AAV9 variant used in this study. The deleted *v-cath* and *ChiA* gene regions in VoyBac1.1 were called the VC locus and included an introduced, unique I-CeuI homing endonuclease (HEN) recognition site. This enabled endonuclease digestion and T4 ligase cloning into the bacmid genome, as was described previously for BEV cloning [26]. The 140 kb VoyBac1.1 bacmid contains a unique FseI restriction endonuclease (REN) site in the *global transactivator* (*gta*) gene ORF and a unique AvrII REN site in the *ecdysosteroid UDP-glycosyltransferase* (*egt*) gene ORF. These sites were exploited for the insertion of foreign genes into the VoyBac1.1 bacmid.

#### 2.3.3. Bacmid VoyBac1.1-LacR-LacOVP1 (VB-LacR-LacOVP1)

Bacmid VoyBac1.1-LacR-LacOVP1 is called VB-LacR-LacOVP1 and is schematically illustrated in Figure 3A. This bacmid construction required two T4 ligase ligation steps, which are described as follows. 

VoyBac1.1 was linearized with the I-CeuI HEN enzyme (#R0699L, New England Biolabs) and then ligated to a synthetically made (GenScript, Piscataway, NJ, USA) I-Ceu-LacO-p10-LacO-VP1-I-CeuI cassette using T4 ligase (#M0202L, New England Biolabs). The Lac LacO-flanked *p10* promoter region for VP1 included 186 bp of the 5′ untranslated region of the *Ac*MNPV *p10* gene, corresponding to 118,726 to 118,906 of the *Ac*MNPV E2 genome (GenBank KM667940.1). The *p10* promoter region was modified to have three internal ATG start codons changed to TTG, corresponding to E2 genome locations 118,774, 118,782 and 118,794. The LacO sequences on either side of the *p10* promoter region were 5′-G A T T G T G A G C G C T C A C A A T T-3′. The LacO-flanked *p10* promoter region was followed by a proprietary ATG translation context sequence and the corresponding VP1 ORF. The VP1 ORF was followed by a 166 bp region that included the Herpes Simplex Virus (HSV) thymidine kinase (TK) polyadenylation signal. The non-palindromic cohesive ends generated by the I-Ceu HEN digest and the I-CeuI-LacO-p10-LacO-VP1-I-CeuI cassette design favored insertion in the same orientation as the upstream *gp64* gene in the bacmid.

The resulting VoyBac1.1-LacOVP1 intermediate bacmid was digested with AvrII REN (#R0174L, New England Biolabs), which linearized the bacmid and disrupted the *egt* ORF. Disrupting the non-essential *egt* ORF does not impair the baculovirus [26]. A synthetically made (GenScript) AvrII-*polh* promoter-LacR ORF-AvrII cassette was ligated using T4 ligase. The cassette included a 92 bp *polh* promoter region upstream of the *LacR* ORF that corresponded to sequence regions 4429 to 4520 of *Ac*MNPV E2 (GenBank KM667940.1). The cassette included a 135 bp SV40 polyadenylation region at the 3′ end of LacR. A VoyBac1.1-LacR-LacOVP1 bacmid clone was selected that had a *polh*-*LacR* cassette in the same orientation as the disrupted *egt* ORF.

#### 2.3.4. Bacmid VoyBac1.1-LacR-Rep-VP3 (VB-LacR-Rep-VP3)

A LacR-Rep-VP3 multi-cassette was made synthetically (GenScript) in a Tn7 transposition donor plasmid for bacmids [23] and was introduced into the VoyBac1.1 bacmid by Tn7 transposition in *E.coli* following previously described methods [23]. The final construct was called VB-LacR-Rep-VP3 and is schematically illustrated in Figure 4A. The *polh* promoter, *Rep* ORF, and *p10* promoter, Cap ORF, regions of this multi-gene cassette had the same configuration as described by Smith et al., 2009 [10]. The only difference was that we deleted 585 bp of the 5′ end of the VP1 ORF, leaving 21 bp of the VP1 ORF sequence upstream of the ATG translational start site for the VP3 ORF. This was carried out to preserve the translational initiation context of VP3. The LacR gene portion (LacR) of the multi-cassette included a hybrid early/late/very late promoter. The early/late part of the promoter was a −166 early/late promoter region from the *Op*MNPV *gp64* gene, as described in Blissard & Rohrmann 1991 [27]. This *Op*MPNV *gp64* promoter was modified with a G-to-C mutation at position −77 to remove a translated minicistron ATG [28]. The *Op*MNPV *gp64* promoter was followed by the spacer sequence G C T A G C G T A T A C and the same 92 bp *polh* promoter LacR ORF construct used in VB-LacR-LacOVP1. A minicistron ORF was identified in the *polh* promoter upstream of the very late ATAAG transcriptional initiation site for polh protein. We used a mutagenic PCR primer (5′-G G C T A G C G T A T A C A T C **T** T G G A G A T A A T T A A A **T** T G A T A A C-3′) to change minicistron ATGs to TTGs and positions −74 and −89 of the *polh* promoter. We refer to this promoter-modified cassette as Opg64polh^mt^LacR and the original design as Opg64polh^wt^LacR. These two Opgp64polh promoter versions were assembled into two VB-LacR-Rep-VP3s for comparative evaluation. 

#### 2.3.5. Bacmid VoyBac1.1-LacOVP1-LacOVP2-LacR-LacR-Rep-VP3 (VB-LacRepCap)

The VoyBac1.1-LacOVP1-LacOVP2-LacR-LacR-Rep-VP3 bacmid is illustrated in Figure 5A and is called VB-LacRepCap. The VB-LacRepCap bacmid has two copies of LacR in different locations to ensure abundant expression of LacR. VB-LacRepCap bacmid combined VB-LacR-LacOVP1 and VB-Lac-Rep-VP3. It also included a LacOVP2 cassette. Construction of VB-LacRepCap was carried out sequentially, starting with VB-LacR-LacOVP1, adding a LacOVP2 cassette by T4 ligase ligation into the FseI REN site, and finally adding the LacR-Rep-VP3 multi-cassette by Tn7 transposition. The LacOVP2 cassette insertion required the unique AvrII REN site in the bacmid. The donor plasmid carrying the LacR-Rep-VP3 multi-cassette had several AvrII REN sites and thus was introduced after LacOVP2 cassette insertion.

The VB-LacR-LacOVP1 bacmid was linearized with the FseI REN enzyme (#R0588L, New England Biolabs) and then ligated to a synthetically made (GenScript) LacO-p10-LacO-VP2 cassette using T4 ligase (#M0202L, New England Biolabs). The LacO-p10-LacO promoter region of the cassette was identical to the corresponding promoter region for LacOVP1. We discovered that the *gta* gene disruption compromised baculovirus replication. The *gta* gene promoter region and the part of the *gta* ORF that was disrupted by FseI REN digestion were restored in the LacO-p10-LacOVP2 cassette by the inclusion of a 487 bp region corresponding to 33,799 to 34,285 in *Ac*MNPV E2 (GenBank KM667940.1) at the 3′ end. The result was a LacO-p10-LacOVP2-restored gta bacmid, and this intermediate bacmid was called VoyBac1.1-LacOVP1-LacOVP2-LacR. It was not used in experiments.

VoyBac1.1-LacOVP1-LacOVP2-LacR bacmid was then modified by Tn7 transposition of the LacR-Rep-VP3 multi-cassette used in the making of VoyBac1.1-LacR-Rep-VP3. The multi-cassette used included the Opg64polh^mt^LacR with the minicistron deletions in the *polh* promoter. The final bacmid construct was called VB-LacRepCap (Figure 5A).

The VB-LacRepCap bacmid was 154,780 bp in size. LacOVP1, LacOVP2, polhLacR (LacR^egt^), and LacR-Rep-VP3 cassettes were separated by 72 kbp, 23 kbp, 15 kbp, and 32 kbp, respectively. There were many baculovirus or bacmid genes between these cassettes. Locally, the LacOVP1 cassette was downstream and commonly oriented with the well-expressed early/late *gp64* gene, and LacOVP2 was downstream and commonly oriented with the weakly expressed (relative to *gp64*) late *Lef12* gene. The LacR^egt^ cassette in the egt locus was in the same orientation as the disrupted *egt* ORF, and the flanking *Lef1* and *ODVe26* genes were pointed away from this LacR cassette. The LacR-Rep-VP3 cassette in the Tn7 locus was surrounded by bacterial genes that would not be actively transcribed in the context of baculovirus infection.

#### 2.3.6. Bacmid VoyBac1.1-ITR-SEAP-ITR (VB-ITR-SEAP)

To make a VoyBac1.1-ITR-SEAP-ITR bacmid, a Tn7L-ITR-SEAP-ITR-Tn7R donor plasmid was synthesized (GenScript) for Tn7 transposition [23] into VoyBac1.1. Inverted terminal repeat (ITR) sequences from the AAV2 serotype flanked a human cytomegalovirus (CMV) promoter (GenBank X03922.1) and an ORF for human secreted embryonic alkaline phosphatase (SEAP) (GenBank NM_001632.5) in regions 53–1568. The promoter included additional proprietary enhancer elements and was 872 bp in length. The 3′ UTR region was 996 bp and included a poly adenylation signal as well as proprietary stuffer sequences, such that the final size of the ITR-SEAP-ITR transgene cassette in the VB-ITR-SEAP bacmid was 3723 bp.

### 2.4. T4 Ligase Cloning Foreign Genes Cassettes into the Bacmid

Gene cassettes for T4 ligation were cut from synthetically made donor plasmids (GenScript) by REN digest and agarose gel purified using the QIAEX II Gel Extraction Kit (#20021, Qiagen, Germantown, MD, USA). Bacmid DNA was purified from *E.coli* and linearized by cutting at corresponding REN sites in the bacmid genome. Bacmid REN digests were heat inactivated at 80 °C for 20 min and combined with gel-purified gene cassettes at 1:10 molar ratios. T4 ligase was added, and ligations were incubated for 3 h at 20 °C. The resulting ligations were ethanol precipitated, suspended in TE buffer, and transformed into electrocompetent NEB 10-beta *E. coli*. Bacmid clones were selected on 50 μg/mL kanamycin-LB agar plates. PCR primers flanking the location of insertion in the bacmid were used to screen for single-copy insertions, and bacmid constructs were sequence verified.

### 2.5. Tn7 Locus Cloning Foreign Gene Cassettes into the Bacmid

Foreign genes were cloned into the attTn7 site (Tn7 locus) by transforming NEB 10-beta *E. coli* with the parental bacmid, the T7 transposase helper plasmid pMON7124 [23], and a foreign gene cassette containing the Tn7L/Tn7R donor plasmid [23]. White colony clones were isolated on LB Agar plates with 7 µg/mL gentamycin, 50 µg/mL kanamycin, 10 µg/mL tetracycline, 100 µg/mL Bluo-gal, and 40 μg/mL Isopropyl β-D-1-thiogalactopyranoside (IPTG) (#L1924, Teknova, Hollister, CA, USA). Bacmid clones were cultured in 50 µg/mL kanamycin media, purified of total DNA, retransformed into *E. coli*, colony isolated on 50 μg/mL kanamycin LB agar plates, and screened for colonies lacking helper plasmid and donor plasmid. Recombinant Tn7 locus bacmids were confirmed using PCR.

### 2.6. Sucrose Cushion rAAV Purification

Twenty-five mL volumes of BEV-infected *Sf9* cells (6.0 × 10^6^ cells/mL) were lysed in culture media by adding 1.25 mL of 10% *w*/*v* triton X-100, 1.25 mL of 2 M arginine, and 1.2 μL of 250 U/μL benzonase (#E1014, Millipore, Burlington, MA, USA). The 50 mL mini bioreactor tubes were placed back at 28 °C and shaken for 2 h at 260 rpm in an orbital shaker. A 300 μL aliquot was set aside for crude lysate rAAV titer determination, and the remaining was centrifuged for 7 min at 7000× *g*. Soluble lysate supernatants were transferred to open-top, 38.5 mL ultracentrifuge tubes (#344058, Beckman Coulter, Indianapolis, IN, USA). Lysates were underlaid with 6 mL of 20% *w*/*w* sucrose in PBS (#P0200, Teknova, Mansfield, MA, USA). Tubes were placed in an SW32 Ti rotor (#369650, Beckman Coulter) and centrifuged at 30,000 rpm (67,214× *g* min to 153,445× *g* max) for 1 h at room temperature in a Beckman Coulter Optima LE80 Ultracentrifuge. The supernatant lysate upper layer and sucrose solution were aspirated, leaving only the sucrose cushion pellet at the bottom of the tube. A total of 500 μL of PBS was added to the pellets, and the pellets were suspended in the PBS, transferred to 1.5 mL cryotubes, and suspended. After suspension, cryotubes were centrifuged for 7 min at 20,000× *g* in a microcentrifuge. The supernatants were collected and transferred to fresh 2.0 mL cryotubes. These supernatants were stored at 4 °C while experiments were conducted and at −20 °C if longer-term storage was needed.

### 2.7. Affinity Purification, CE-SDS Analysis, and Empty/Full Ratios

Eight hundred mL cultures of BEV-infected *Sf9* cells were lysed in culture media supplemented with 0.5% *v*/*v* triton X-100, 200 μM arginine, and benzonase, as described earlier. Cell lysate rAAV capsids were affinity captured using POROS^TM^ Capture Select^TM^ AAV9 affinity resin (#A27354, Thermo Fisher Scientific). Resin-captured rAAV were washed with 20 mM Tris, 1 M NaCl (pH 8.0), bridge washed with 50 mM Na_2_PO_4_, 350 mM NaCl (pH 5.5), and then eluted with 200 mM glycine, 50 mM NaCl (pH 3.0). Eluted rAAV were neutralized by adding Tris-base to the 150 mM final concentration. All steps in purification contained 0.001% *v*/*v* pluronic F-68 (#24040032, Thermo Fisher Scientific). Capillary electrophoresis of sodium dodecyl sulfate (CE-SDS) was conducted using a Sciex PA-800 capillary electrophoresis system. Affinity chromatography-purified rAAV capsid samples were diluted in water to a protein concentration of 100 μg/mL. Quantification of separated proteins was carried out by measuring peak area absorbances at 220 nm, which were normalized to the molecular weight to accurately calculate ratios of VP1, VP2, and VP3. Empty-full ratios of capsids were determined using size exclusion chromatography multiangle light scattering (SEC-MALS) as described [29].

### 2.8. Western Blots

For total cell lysate Western blots, *Sf9* cells were collected by centrifugation for 2 min at 20,000× *g*, and then cell pellets were resuspended at 2.0 × 10^3^ cells/μL in PBS, pH 7.4 (#0010072, Thermo Fisher Scientific). NuPAGE™ LDS Sample Buffer (#NP0007, Thermo Fisher Scientific) and 10× NuPAGE™ Sample Reducing Agent (#NP0004, Thermo Fisher Scientific) were combined to 2× concentrations in PBS, pH 7.4. Resulting 2× LDS Sample Buffer/Reducing Agent were combined with equal volumes of PBS-diluted cells. For sucrose cushion sample Western blots, 40 μL of PBS, pH 7.4, solubilized sucrose cushion pellets were diluted with 15 μL of 4× LDS Sample Buffer and 6 μL of 10× Sample Reducing Agent. The final sample represented material generated from 2.0 × 10^5^ *Sf9* cells/μL. All SDS-PAGE samples were suspended and then heat denatured for 10 min at 80 °C. Volumes of 10 μL were loaded onto NuPAGE™ 4–12% Bis-Tris 1.0 mm × 17 well SDS-PAGE gels (#NP0329BOX, Thermo Fisher Scientific). Proteins were fractionated by electrophoresis in 1× MOPS buffer (#BP-178, Boston BioProducts, Milford, MA) for 90 min at 120 V and then transferred to nitrocellulose membranes (#1704159, Bio-Rad, Hercules, CA, USA). Blots were blocked for 1 h with 5% *w*/*v* Blotting-Grade Blocker (#1706404, Bio-Rad) diluted in 1× TBS-T (#IBB-180X, Boston BioProducts). Primary mouse monoclonal antibodies were diluted to 1:2000 in TBS-T and incubated with blocked blots for 90 min. The anti-Lac Repressor antibody used was mouse monoclonal antibody Anti-LacI [9A5] (#ab33832, Abcam, Cambridge, MA, USA). The anti-Capsid antibody used was the mouse monoclonal antibody, anti-AAV VP1/VP2/VP3 antibody (B1) (#03-61058, American Research Products, Waltham, MA, USA). Secondary goat anti-mouse horse radish peroxidase (HRP) conjugated antibodies were diluted 1:10,000 in TBS-T and incubated for 90 min with primary antibody probed blots after they had been washed with 2× TBS-T three times. The secondary antibody used for all Enhanced Chemiluminescence (ECL) Western blots was a HRP-conjugated goat anti-mouse IgG H&L antibody (Abcam, Goat Anti-Mouse IgG H&L (#ab6789, Abcam). Western blot signals were detected using Clarity Western ECL Substrate (#170-5060, Bio-Rad) and an Azure Imager c300 (Azure Biosystems, Dublin, CA, USA). All images were collected as Tiff files and analyzed using ImageJ software [30].

### 2.9. rAAV Transduction Assays

HEK 293T cells were seeded into 96-well plates at 5.0 × 10^4^ cells/well in 80 μL of Opti-MEM™ Media (#51985091, Thermo Fisher Scientific) supplemented with 5% *v*/*v* fetal calf serum. After 2 days of incubation at 37 °C in 5% *v*/*v* CO_2_, cells were transduced with rAAV-containing sucrose cushion samples. In a U-bottom 96-well plate, 20 μL volumes of 1/10, 1/20, 1/40, and 1/80 diluted AAV samples in PBS were combined with 200 μL of Opti-MEM media and 8 uL of 100× Antibiotic-Antimycotic (#15240112, Thermo Fisher Scientific). Thirty-five μL volumes of these rAAV dilutions were transferred to the wells of the HEK 293T cell plates. This was conducted in triplicate for each rAAV sample.

### 2.10. Determination of rAAV Titer by Q-PCR

Packaged genome copies of transgenes were determined by Q-PCR as described previously [21].

### 2.11. SEAP Activity Assay Assays

AAV samples were added to 96-well plates of freshly seeded HEK 293T cells. At 4 days post-transduction with rAAV, the media from HEK 293T cells was transferred from 96-well tissue culture plates to 96-well PCR plates that were sealed with aluminum covers and heated at 65 °C for 15 min to inactivate cellular alkaline phosphatases. Fifty μL volumes of heat-treated media were transferred into 96-well microtiter plates and combined with 50 μL volumes of p-nitrophenylphosphate BioFX AP-Yellow One (#NC9444916, Thermo Fisher Scientific). Samples were incubated for 24 h at room temperature in the dark to reach the endpoint. The optical density at 405 nm (OD405) was read using a Synergy HTX microplate reader (BioTek, Winooski, VT, USA). Every assay included a recombinant shrimp alkaline phosphatase (#M0371S, New England Biolabs) standard that had been serially diluted in Opti-MEM media and not heat-treated. Shrimp alkaline phosphatase was confirmed by the same assay to have equal unit phosphatase activity as calf intestinal alkaline phosphatase (#18009027, Thermo Fisher Scientific).

### 2.12. Statistical Calculations

Standard deviations for Q-PCR-derived rAAV genome titers or SEAP activity assays were calculated from the average of three repeated samples. Standard deviations for potency (SEAP activity/rAAV genome) were calculated from the square root of the sum of the squared coefficients of variation of Q-PCR titer and SEAP activity.

## 3. Results

### 3.1. LacR Was Confirmed to Be Able to Regulate Expression of a LacOVP1 Construct in the Context of Baculovirus Infection

The goal of this work was to make a LacR-inducible system for regulating AAV capsid expression in the context of baculovirus-infected insect cells. To our knowledge, LacR has not been reported to regulate very late hyperexpressed *p10* or *polh* baculovirus promoters in the context of baculovirus infection. In Slack & Blissard, 1997 [22], LacR was used to regulate transcription of plasmids transiently transfected into *Sf9* cells and was never applied to a strong baculovirus promoter in the context of baculovirus replication. An intermediate recombinant bacmid construct called VB-LacR-*LacO*VP1 (Figure 3A) was made to confirm the ability of LacR to regulate the hyperexpressed very late baculovirus *p10* promoter. The LacR protein is a tetramer that binds to two LacO sequences simultaneously [31]. We placed LacO sequences on either side of the very late baculovirus *p10* promoter, driving expression of VP1 (Figure 3B).

The VB-LacR-*LacO*VP1 bacmid was evaluated under different IPTG inducer concentrations. We observed a gradient of VP1 expression proportional to the IPTG concentration (Figure 3D), and the range of LacR regulation of VP1 was about 50% based on estimates of the ECL Western blot signal (Figure 3F).

In Western blots, it was noticed that there was a 60 kDa protein in addition to the expected 82.1 kDa VP1 protein (Figure 3B). This corresponded to the 60.6 kDa VP3 protein, and it could be from leaky translational scanning of the VP1 mRNA transcript. The 60 kDa protein abundance had the same response to IPTG as did VP1. An unexpected observation was to see LacR protein abundance correlate inversely with IPTG concentration. This could be explained by competition for translation as IPTG induction released LacR transcriptional repression of the *LacO*-p10-*LacO* promoter controlling VP1 expression. This observation prompted us to design a second copy of LacR in the bacmid with an earlier baculovirus promoter as well as a very late baculovirus promoter to ensure LacR was always present in abundance.

### 3.2. Development of a Hybrid Early/Late/Very Late Opgp64polh Promoter for Expressing LacR and Evaluation in Intermediate Bacmid VB-LacR-Rep-VP3

Our experiments with VB-LacR-*LacO*VP1 revealed that *LacR* expression from a *polh* promoter was being squelched by selfish transcription from LacO-P10-LacO promoters driving VP1 expression when IPTG induced LacR (Figure 3). Baculovirus gene expression has a distinct temporal cascade of early (6 h), late (12 h), and hyper-expressed very late gene expression (24 h). We decided to express LacR from an earlier baculovirus promoter, which would not compete with the very late baculovirus promoters driving AAV capsid VP expression. We also did not want to lose the abundant *polh*-promoter-driven expression of LacR. A hybrid *Op*MNPV *gp64*-*polh* promoter was designed to drive LacR expression, and the promoter was called *Opgp64polh*. A 166 bp early and late promoter from the *Op*MNPV baculovirus *gp64* gene [23] was cloned upstream of a *polh* promoter (Figure 4C). We tested the *Opgp64polh* promoter in the bacmid construct VB-LacR-Rep-VP3 (Figure 4A). Initially, we did not see earlier expression of LacR, and we subsequently identified a minicistron ORF between the Opgp64 and *polh* promoters in the *Opgp64polh*LacR cassette (Figure 4D). We used PCR mutagenesis to remove the minicistron translational start sites in the *polh* promoter. This resulted in the desired earlier expression of LacR (Figure 4G) relative to VP3 expression (Figure 4F). We elected the *Opgp64polh^mt^*LacR cassette (Figure 4E) in VB-LacR-Rep-VP3 and in the following progeny bacmid, VB-LacRepCap (Figure 5A). 

### 3.3. Final Bacmid VB-LacRepCap and Initial Time Course Analysis 

A final bacmid construct for this study called VB-LacRepCap (Figure 5A) was made, which combined all the elements of the previously described bacmids and added a Lac-inducible VP2 ORF in the *gta* locus of the bacmid (See Section 2.3.5). VB-LacRepCap retained two copies of LacR to ensure the system was saturated with LacR. 

The first experiment with bacmid VB-LacRepCap was a time-course experiment to evaluate LacR and capsid VP protein production (Figure 5D). The goal of the experiment was to determine the extent of LacR repression of VP1 and VP2 expression over the life cycle of baculovirus infection. We observed LacR expression at 14 hpi preceding the expression of VP1, VP2, and VP3 capsids. This would be expected, as one copy of LacR has an early/late/very late promoter. However, between 14 hpi and 24 hpi, VP1 was more abundant than VP2 and similar in abundance to VP3 (Figure 5E). The *LacO*-p10-*LacO* very late promoter of VP1 should have had the same temporal expression profile as VP2. We speculate that more abundant early VP1 translation may be from leaky scanning of upstream baculovirus *gp64* gene mRNA transcripts, which would not be affected by LacR. The LacOVP2 gene was cloned downstream of the baculovirus *Lef12* gene promoter, which is 50-fold less transcribed than the *gp64* gene [32]. When we alternated the locations of LacOVP1 and LacOVP2, there was more VP2 expression than VP1. We chose to continue working with the first construct design with the acknowledgement that the cumulative capsid ratios seen in Western blots had imperfect homogeneity at the beginning of the baculovirus infection cycle. The vast majority of VP1 and VP2 transcripts would come at very late times from the *p10* promoters, which are expressed at much higher levels than *gp64*.

### 3.4. There Is Competition among Very Late Promoters as IPTG Induces LacR Repression of LacO Regulated Very Late Promoters 

Once we had established the maximal repression of VP1 and VP2 expression by LacR in the context of VP-LacRepCap, we investigated the IPTG induction at very late times post-infection when rAAV are typically expressed and harvested (Figure 6). As with VB-LacR-LacOVP1, the responsive concentrations of IPTG were between 0 μM and 50 μM. Cumulative VP1 and VP2 abundances increased proportionally in response to IPTG induction of LacR repression of *LacO*-p10-*LacO* promoters. Estimated VP1:VP2:VP3 capsid ratios in infected *Sf9* cells ranged from 15:17:68 at 0 μM IPTG to 30:32:38 at 50 μM and 100 μM IPTG. LacR was not able to repress VP1 and VP2 expression enough to obtain an ideal 1:1:10 ratio. When VP1 and VP2 became more abundant, there was an unexpected reciprocal drop in VP3 abundance. This can be explained by the *LacO*-p10-*LacO* promoters for VP1 and VP2, drawing away very late transcription factors from the *p10* promoter and driving expression of VP3.

### 3.5. Increasing Abundances of VP1 and VP2 Led to Reduced Capsid Titers at Three ITR-SEAP:LacRepCap BEV Co-Infection Ratios 

After determining IPTG concentrations for the regulation of capsid ratios, we then optimized co-infection ratios for the VB-LacRepCap with a SEAP transgene-carrying bacmid called VB-ITR-SEAP (See Section 2.3.6). *Sf9* cells were co-infected with VB-ITR-SEAP and VB-LacRepCap Bacmids at co-infection ratios ranging from 9:1 to 1:12. In these 25 mL scale mini bioreactor shake flask experiments, initial infection MOIs ranged from 0.01 to 0.12 TCID50 units per cell, with, for example, the 9:1 ratio co-infection receiving 0.09 TCID50 units of VB-ITR-SEAP bacmid and 0.01 TCID50 units of VB-LacRepCap bacmid per cell. Low MOI infections were conducted to better emulate larger-scale production of rAAV in the BEV *Sf9* system. In our experience, ITR-transgene bacmid to RepCap bacmid co-infections were optimal at a 3:1 ratio. It was surprising to find that the 3:1 co-infection ratio of VB-ITR-SEAP:VB-LacRepCap bacmids produced a low titer rAAV compared to ratios as high as 1:12 (Figure 7B). It is possible that the VB-LacRepCap bacmid was replicating more slowly, but it had similar viral growth to the conventional RepCap bacmid we were routinely using. Another explanation for needing more of the VB-LacRepCap bacmid could be a reduced abundance of Rep protein in the VB-LacRepCap bacmid due to LacR and capsid VP expression drawing away very late baculovirus transcription factors from the *polh* promoter driving Rep expression. We did not observe by Western blot analysis a deficiency of Rep expression by the VB-LacRepCap bacmid compared to Rep expression from the conventional RepCap bacmid. Higher VB-ITR-SEAP:VB-LacRepCap bacmid co-infection ratios of 9:1, 6:1, and 3:1 also produced more abundant VP1 expression relative to VP2 expression. This resembled the overabundance of VP1 that was observed in the time-course experiment of VB-LacRepCap bacmid at 18 hpi and 22 hpi (Figure 5). In this case, the excessive VB-ITR-SEAP bacmid co-infecting *Sf9* cells with the VB-LacRepCap bacmid was providing more early and late transcription factors to express the baculovirus *gp64* gene upstream of *LacO*VP1, which we have speculated has translatable transcripts of VP1. Despite the imbalance in VP1 and VP2 at high co-infection ratios, there was still some Lac-regulated control of VP ratios. Under all co-infection conditions, rAAV titers were higher when VP1 and VP2 expression were repressed by LacR.

### 3.6. A High Throughput Method Was Developed to Purify rAAV from Bacmid Infected Sf9 Cells 

This inducible system required optimizing both IPTG concentrations and VB-ITR-SEAP:VB-LacRepCap bacmid co-infection ratios for rAAV potency assays on HEK 293T cells. Screening of IPTG concentrations and co-infection conditions often required 25 to 30 samples per experiment. A simple, faster, and reproducible method of rAAV purification was employed based on a single-step sucrose cushion ultracentrifugation method [33]. We achieved rapid and consistent recovery of pelleted rAAV particles through 20% sucrose after only 1 h of ultracentrifugation. The recovered soluble rAAV in PBS-resuspended sucrose cushion pellets were found to be suitable for HEK 293T cell transduction assays and did not require buffer exchange. The sucrose cushion purification should have eliminated monomeric non-assembled capsid proteins, and the 20,000× *g* centrifugation after sucrose cushion pellet resuspension should have eliminated capsid protein aggregates. This method allowed 30 samples to be processed in one day on one centrifuge. Control empty capsids were found to purify through 20% sucrose cushions in similar abundance to DNA-containing capsids produced in the presence of ITR-SEAP transgenes. Thus, there was no discrimination between empty and full capsids in this purification method. A disadvantage of this method of AAV capsid purification was that only about 1% of total rAAV capsids in cell lysates were being recovered. Despite this inefficiency, there were more than sufficient rAAV particles to carry out several 96-well plate HEK 293T transduction assays. 

One additional issue we had with purifying rAAV capsids only by sucrose cushion was that baculovirus capsids were co-purified. When we used these samples to transduce HEK 293T cells, there was SEAP enzyme background activity from the VB-ITR-SEAP bacmid alone control group, which was not detected in the VB-LacRepCap bacmid alone control group. Initially, we thought the background SEAP activity was coming from baculovirus virion transduction of HEK 239T cells. However, we could not detect the baculovirus envelope fusion protein GP64 by Western blot, and the background SEAP activity did not diminish after lysate treatments with higher concentrations of detergents. We discovered that there was SEAP expression by the VB-ITR-SEAP bacmid in *Sf9* cells, and the resulting SEAP enzyme may be associated with baculovirus capsids co-purifying with AAV capsids through sucrose cushions. The baculovirus capsid protein ORF1629 has an affinity for phosphatase enzymes [34]. The VB-ITR-SEAP bacmid alone control preparation SEAP activities were subtracted as residual SEAP background from all VB-ITR-SEAP:VB-LacRepCap bacmid co-infection groups. The background residual SEAP activity was not present when rAAV was purified by immunoaffinity purification and baculovirus capsids were eliminated.

### 3.7. The Yield of Assembled Capsids Decreased as VP1 and VP2 Expression Was Increased Relative to VP3

The VB-ITR-SEAP:VB-LacRepCap bacmid co-infection ratios of 1:1, 1:3, and 1:6 were selected for further optimization with IPTG concentration. Twenty-five mL-scale co-infections of *Sf9* cells were conducted at eight different IPTG concentrations ranging from 0 μM to 200 μM, and the rAAV capsids were purified by 20% sucrose cushion from the cell lysates of baculovirus-infected insect cells. Western blots were used to detect total expressed VP1, VP2, and VP3 capsid proteins in cell lysates and what was recovered as assembled capsids in sucrose cushion pellets (Figure 7). Total capsid proteins present in cell lysates only changed marginally as IPTG concentrations went up. The 1:1 co-infection ratio cell lysate had a lower total capsid protein yield relative to the 1:3 and 1:6 co-infection ratios. Total capsid protein abundance in sucrose cushion purified samples was more affected by varying IPTG concentrations. For all three co-infection groups, increased expression of VP1 and VP2 relative to VP3 reduced the yield of capsid proteins passing through the sucrose cushion. We interpret this as assembled capsids passing through the sucrose cushion and soluble proteins remaining in solution above the sucrose cushion. The capsid ratios in cell lysates differed from the capsid ratios in corresponding sucrose cushion samples. (Figure 8E,F). Interestingly, the abundance of VP1 in sucrose cushion samples relative to VP2 and VP3 was more constant over the 0 to 200 μM IPTG range compared to the cell lysate samples. There appears to be a limit to the amount of VP1 that can be incorporated into assembled rAAV capsids and less of a limit for VP2 incorporation into capsids. 

### 3.8. The Potency of rAAV Capsids Was Tunable Using the Lac Inducible System for Small Scale Production

Potency was determined by transducing HEK 293T cells with rAAV capsids containing the ITR-SEAP reporter transgene and measuring the SEAP enzymatic activity relative to the rAAV dosage on the cells. Before doing potency transduction assays, we measured the rAAV titers from sucrose cushion-purified rAAV (Figure 9A). There was a drop in the rAAV titer of sucrose cushion-purified capsids with increased IPTG concentrations. This was likely due to the lower abundance of capsid proteins in sucrose cushion samples as IPTG concentrations increased (Figure 8D). The potency of recovered capsids did not follow capsid abundance and capsid titer, which both peaked at 0 μM IPTG. Instead, potency was highest between 1 μM and 2 μM IPTG for all three co-infection groups (Figure 9B). Potency dropped by sixfold at 0 μM IPTG and at 5 μM IPTG. Western blots did not show significant differences in VP ratios between 0 μM and 5 μM IPTG. It is possible that there were different ratios over the baculovirus infection cycle that were not observable at the cumulative final harvest time point.

### 3.9. Larger Scale Production of rAAV in Sf9 Cells Confirmed That There Was Limited Incorporation of VP1 into Capsids 

The data presented thus far was generated from sucrose cushion-purified rAAV samples generated at a small 25 mL scale. Production of rAAV for drug therapeutics involves larger-scale *Sf9* cultures and affinity chromatography-based purification [35]. We scaled up to 800 mL *Sf9* cultures and co-infected these cells with ITR-SEAP and LacRepCap BEVs at a 1:6 ratio (0.1:0.6 MOI). These co-infections were conducted at four IPTG concentrations: 0 μM, 2 μM, 10 μM, and 50 μM IPTG. *Sf9* cell lysates made at this time were subjected to affinity chromatography to purify rAAV capsids. A portion of those affinity-purified capsids were further purified by sucrose cushion ultracentrifugation. Expressed capsid VP ratios in cell lysates were determined by Western blot (Figure 10A) or by CE-SDS for both affinity-purified capsids (Figure 10B) and affinity/sucrose cushion-purified capsids (Figure 10C). The most noticeable trend was the nearly constant proportion of VP1 in both affinity and affinity/sucrose cushion purified rAAV capsids, regardless of the IPTG concentration. In contrast, the relative abundance of VP2 in purified capsids increased with IPTG concentration, just as it did in cell lysates. These data corroborate with 25 mL scale data and show that capsid assembly is not stochastic with a limit to the amount of VP1 that can be incorporated into capsids.

### 3.10. Larger Scale Production Did Not Show Expected Optimal Potency at 2 μM IPTG

We did not see the expected increased potency when capsids were produced in the presence of 2 μM IPTG, as was observed in smaller 25 mL-scale experiments (Figure 10B). Instead, rAAV potency on HEK 293T cells was highest at 0 μM IPTG (Figure 11A,B). The percent of capsids containing ITR-SEAP transgenes in affinity purified samples was measured using SEC-MALS and was found to be highest at 0 μM IPTG and lowest at 50 μM IPTG (Figure 11C).

## 4. Discussion

The *E. coli* Lac repressor (LacR) was selected in this study as a regulator of VP1 and VP2 expression due to its success in other eukaryotic virus platforms such as Vaccinia [36,37,38] and adenoviruses [39]. LacR is a component of the *E. coli* Lac Operon that was the first described inducible regulatory system [40] and which has been well characterized (For review, see Lewis 2005 [41]). LacR was previously shown to be functional in insect cells and was allosterically regulated (induced) by IPTG [22,42].

In nature, VP1, VP2, and VP3 evolved to be overlapping ORFs due to the restricted genome capacity of AAV. These overlapping ORFs were emulated when researchers first cloned rAAV constructs into a bacmid-based BEV system [9,10]. Bacmids do not have a restricted genome size like AAV. Capsid protein VP1, VP2, and VP3 ORFs were separately cloned and stably expressed from three different loci in the bacmid. We also cloned and expressed the *E. coli LacR* gene into another unique locus in the bacmid. This modular recombinant bacmid design was stable and allowed for independent modifications of the various elements. Engineering the baculovirus *p10* promoters of VP1 and VP2 genes with double LacO’s enabled IPTG-inducible LacR regulation. This is the first report of a very late “hyperexpressed” baculovirus promoter being placed under inducible regulation. Due to the strength of *p10* transcription, the repression of VP1 and VP2 expression was not complete. Repression was sufficient for the intended purpose of regulating the expression of VP1 and VP2 relative to VP3 for successful AAV capsid assembly.

IPTG induction of LacR repression of VP1 and VP2 expression relative to VP3 was tunable such that the potency of the resulting capsids could be optimized in small 25 mL-scale *Sf9* cultures. This did not translate to larger-scale 800 mL production, with the non-induced LacR repression producing the most potent rAAV. A possible reason for this was that the VB-ITR-SEAP bacmid and VB-RepLacCap bacmid co-infection ratios at a larger scale needed to be further optimized. As shown in Figure 9B, having too little VB-LacRepCap bacmid relative to VB-ITR-SEAP bacmid reduced the ability to tune potency with IPTG. The 800 mL-scale VB-ITR-SEAP and VB-LacRepCap bacmid co-infections were also carried out with an MOI about 100-fold lower than the 25 mL-scale co-infections. This lower MOI may have generated larger populations of infected *Sf9* cells with suboptimal co-infection ratios based on the normal distribution of two co-infecting bacmids [43]. 

From this work, it can be concluded that there was a decline in total assembled capsids and rAAV titers as expressed VP1 and VP2 abundances were increased relative to VP3. Our data agree with the findings of Gao et al., 2014 [44] which showed VP1 and VP2 abundances influenced capsid yield. We saw this trend of expressed VP1 and VP2 abundance affecting yield to be consistent with repeated experiments and with other rAAV9 capsid variants. Our results contradict the findings of Bosma et al., 2018 [17] and show a decline in the percent of full capsids resulting from increased expression of VP1 and VP2 (Figure 10C). A major difference in that study was that it used a rAAV5 serotype, which does not require expression of the frame-shifted AAP protein found nestled in the common sequences of VP1 and VP2 ORFs.

Both VP1 and VP2 ORFs have complete copies of the AAP ORF. We did not address how modulating VP1 and VP2 expression affected AAP expression. The canonical ATG start codons for VP1 and VP2 would likely lead to less translational leaky scanning and, thus, less translation from the downstream non-canonical CTG start codon for AAP. However, as VP1 and VP2 transcription increased with IPTG induction, there would be more AAP produced because of translational leaky scanning.

Capsid ratio analysis methods like CE-SDS or Western blot used here show only the mean distribution of VP1, VP2, and VP3 in purified capsids. The VP ratios found in individual capsids are a random normal distribution of VP1, VP2, and VP3, dependent on the starting abundances of VPs during assembly [45]. To complicate this further, the BEV in this Lac-inducible system did not offer a steady-state relative expression of VP1, VP2, and VP3 due to transcriptional interference from surrounding baculovirus genes in the separated cloning loci. 

Our data show that there is an upper limit to the amount of VP1 that can be included in assembled capsids. This is best illustrated in Figure 8, where the crude cell lysate abundance of VP1 was 20% at 50 μM IPTG and was only 9% in the affinity/sucrose cushion purified capsids. At the same time, VP2 abundance in the cell lysate was 31% and was 27% in the affinity/sucrose cushion purified capsids. It is concluded that assembled capsids tolerate a higher abundance of VP2 compared to VP1. Overabundance of VP1 leads to reduced levels of capsid assembly and reduced potency. We have also previously observed that deficiency of VP1 in capsids does not affect capsid assembly or titer but does reduce capsid potency. Too much or too little VP1 reduces potency.

It has been suggested that IPTG should not be used in processes to make human therapeutics because it is too costly and toxic to humans [46]. Kaneda et al., 1998 [47] reported IPTG to be nontoxic to Hela cells at 5000 μM; Cronin et al., 2001 [48] found IPTG toxicity in mice did not occur until concentrations were at 20,000 μM; and Figge et al., 1988 [49] reported toxicity of IPTG to monkey cell lines only when concentrations reached 50,000 μM. In the current study, we found IPTG to be well tolerated by *Sf9* cells and were able to culture *Sf9* cells at IPTG concentrations as high as 18,000 μM IPTG with no effect on cell growth. This concentration of IPTG was 1000-fold higher than the working IPTG concentration range needed in our inducible system. In addition, after rAAV virions are purified from baculovirus-infected insect cell lysates by affinity chromatography, IPTG would not be expected to be present at significant concentrations in the final drug product. At the time of this study, the cost of the ESF-AF insect cell culture medium was $62/L. Animal-free, high-purity IPTG (mw 238.3) costs $205 for 5 g (CAS 367-93-1 Calbiochem). Even at the highest foreseeable concentration of 100 μM IPTG, we would need 24 mg of IPTG/L, costing $0.98/L, a 1.6% increase in media cost. The optimal IPTG concentration for rAAV potency in this study was 2 μM IPTG, thus cutting the IPTG cost to 0.03% of the media cost. IPTG is not too costly or toxic at the working concentrations for this system.

The purpose of this work was to improve the potency of rAAV9 serotype capsids produced in the *Sf9*/BEV system. While we demonstrated repression of VP1 and VP2 expression via LacR regulation, further refinement of the transcriptional elements may enable LacR regulation that is more viable and tunable for rAAV. We also desired a better understanding of how AAV capsid protein ratios affected rAAV. This was achieved, and we gained a better understanding of the semi-stochastic nature of AAV assembly.

## Figures and Tables

**Figure 1 viruses-16-00051-f001:**
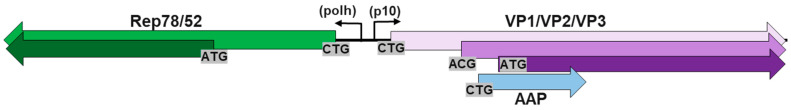
An Original and Commonly Used rBEV (Bacmid) RepCap Dual Cassette Design for Producing rAAV *Sf9* Cells. The overlapping Rep78/Rep52 and Cap-related VP1/VP2/VP3/AAP genes are shown in context with the bipartite baculovirus very late *polh*/*p10* promoter driving expression of Rep and Cap. This 4389 bp dual cassette is cloned into a single location in the BEV. Non-canonical translation start-site sequences are indicated. Rep genes Rep78 (1866 bp) and Rep52 (1194 bp) have common ORFs and 3′ regions. Cap genes VP1 (2111 bp), VP2 (1800 bp), and VP3 (1605 bp) have common ORF 3′ regions. The AAP gene ORF is out of frame from Cap genes, begins 526 bp from the VP1 translational initiation site, and is 594 bp. Only the VP1 and VP2 ORFs overlap the AAP ORF.

**Figure 2 viruses-16-00051-f002:**
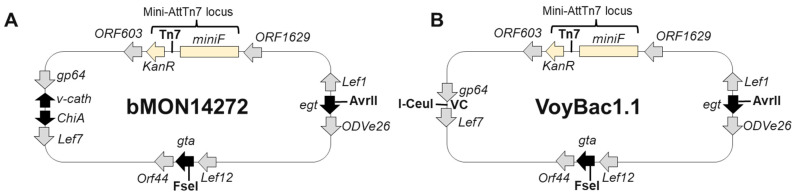
Comparison of the Original bMON14272 Bacmid and VoyBac1.1 Bacmid Backbones. The 142,272 bp pMON14272 bacmid (**A**) and the 140,505 bp Voyager VoyBac1.1 bacmid (**B**) are schematically illustrated. Both bacmids have the same min-AttnTn7 transposition elements and a 6619-bp miniF replicon. The location of Tn7 transposition insertion is indicated by Tn7. The *v-cath* and *ChiA* deletion regions in VoyBac1.1 are indicated as VC. Baculovirus genes surrounding Tn7, FseI, AvrII, and I-CeuI sites and their directions of transcription are also illustrated.

**Figure 3 viruses-16-00051-f003:**
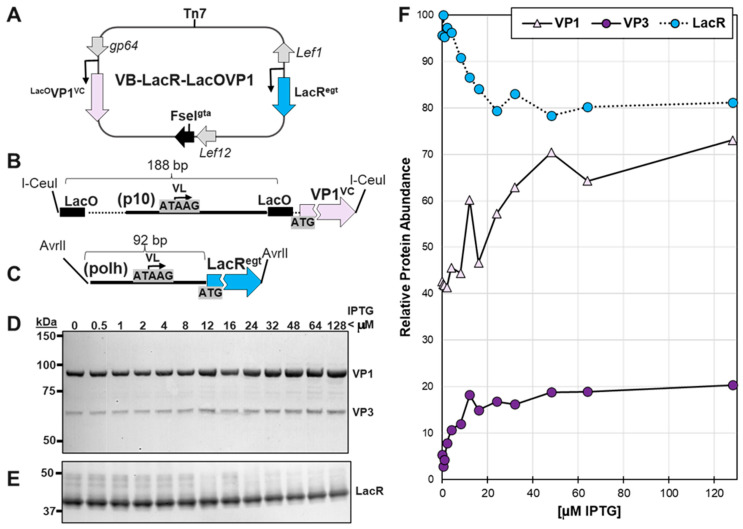
Testing LacR Regulation of LacOVP1 Expression in the Context of Baculovirus Infection. Bacmid VB-LacR-LacOVP1 is schematically illustrated (**A**). The location and transcription directions of baculovirus genes *gp64*, *gta*, and *Lef1* are shown. This bacmid included a LacO-p10-LacO promoter VP1 cassette (**B**) in the VC locus and a *polh* promoter LacR cassette in the *egt* locus (**C**). *Sf9* cells were infected at 10 MOI in the presence of different concentrations of IPTG. At 72 hpi, infected *Sf9* cells were collected, fractionated by SDS-PAGE, and then Western blot probed with an anti-capsid monoclonal antibody (**D**) and an anti-lac repressor antibody (**E**). Relative abundances of ECL-detected proteins were quantified using ImageJ software (version 1.53) and plotted (**F**).

**Figure 4 viruses-16-00051-f004:**
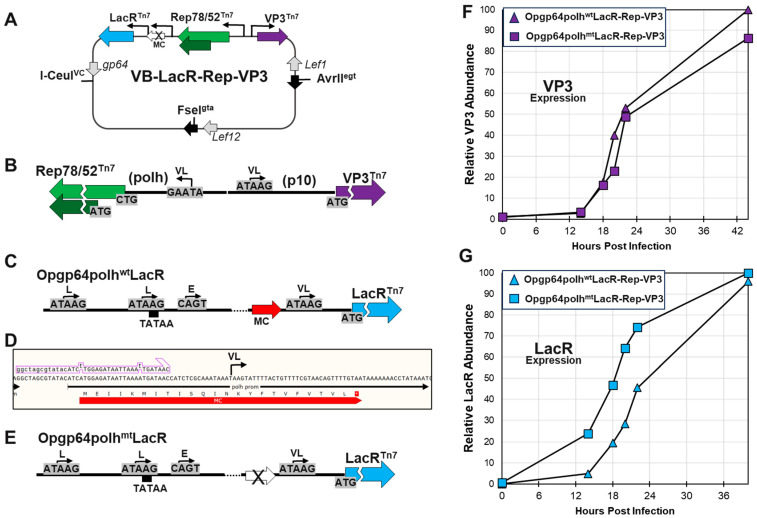
Testing bacmid VB-LacR-Rep-VP3 and Validating a Hybrid Early/Late/Very Late Promoter for LacR Expression. The bacmid VB-LacR-Rep-VP3 is schematically illustrated (**A**). This bacmid was based on VoyBac1.1 and had a LacR-Rep-VP3 multi-cassette inserted into the Tn7 locus. The Rep and VP3 were in opposing orientations and had *polh* and *p10* promoters, respectively (**B**). The LacR ORF was downstream of Rep and in the same orientation. The original promoter design for LacR in this multi-cassette was a hybrid *Opgp64*-*polh* promoter (**C**). This promoter had two late ATAAG transcriptional initiation sites: an early CAGT transcriptional initiation site in the *Opgp64* promoter and a very late ATAAG transcriptional initiation site in the *polh* promoter. A minicistron was present in the *polh* promoter upstream of the *polh* ATAAG transcriptional initiation site (**D**). A mutagenic PCR primer was designed to remove two ATG translational initiation sites in the minicistron (**B**). A mutagenic PCR primer was used to remove the minicistron translational initiation sites from the *polh* promoter, resulting in an Opgp64polh^mt^LacR cassette (**E**). Two versions of VB-LacR-Rep-VP3 were made: Opgp64polh^wt^LacR-Rep-VP3 (original version) and Opgp64polh^mt^LacR-Rep-VP3 (minicistron mutated version). *Sf9* cells were infected at 10 MOI with the two versions of bacmid VB-LacR-Rep-VP3, and insect cells were collected at different hours post-infection. Cell proteins were fractionated by SDS-PAGE, and then Western blots were probed with anti-Cap or anti-LacR antibodies. The relative abundances of Western-detected VP3 (**F**) and LacR (**G**) are plotted on the graphs.

**Figure 5 viruses-16-00051-f005:**
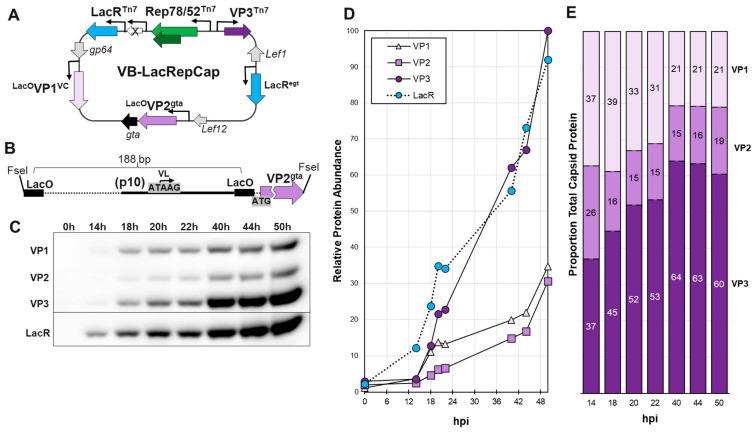
Bacmid VB-LacRepCap and Time Course of Cap and Lac Expression in Sf9 Cells. The bacmid VB-LacRepCap is illustrated schematically (**A**). This bacmid was a combination of the bacmids VB-LacR-VP1 and VB-LacR-Rep-VP3. VB-LacRepCap also included a new LacOVP2 cassette (**B**) in the *gta* locus. *Sf9* cells were infected at 10 MOI with VB-LacRepCap. Cells were collected at different hours post-infection, from 0 hpi to 50 hpi. Cell proteins were fractionated by SDS-PAGE and then Western blot probed with anti-capsid monoclonal antibody and anti-lac repressor antibody (**C**). (**D**) ECL signals of Western-detected proteins were quantified using ImageJ software. (**E**) Relative capsid VP ratios are expressed in bar graph format.

**Figure 6 viruses-16-00051-f006:**
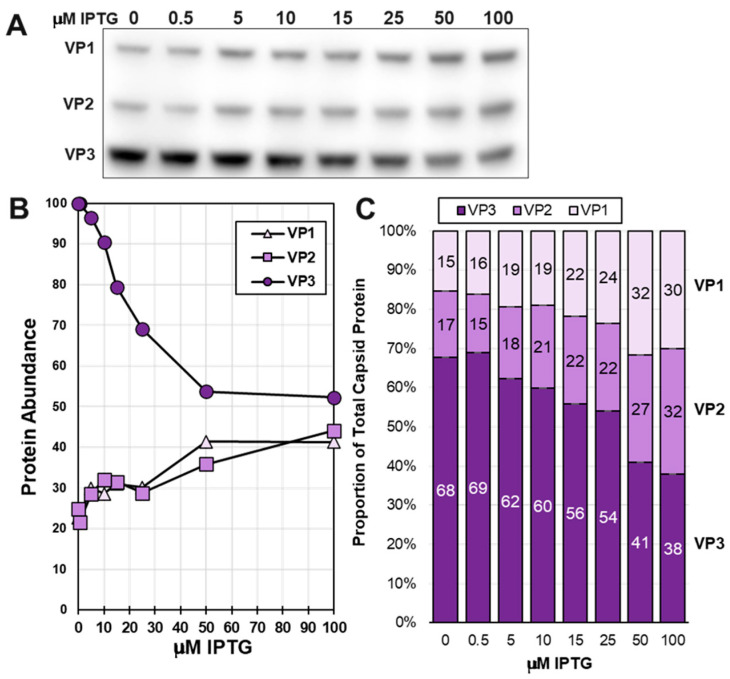
Titration of IPTG Induction of VP1 and VP2 expression. *Sf9* cells were infected at 10 MOI with VB-LacRepCap bacmid in the presence of different concentrations of IPTG. Cells were collected at 72 hpi, and their proteins were fractionated by SDS-PAGE and Western blot probed with an anti-Capsid monoclonal antibody. Relative capsid abundances in infected cells were observed by Western blot (**A**) and ECL signals, which were quantified using ImageJ software. Relative VP1, VP2, and VP3 capsid abundances in infected cells are graphed relative to IPTG concentration (**B**). Calculated capsid ratios are expressed in bar graph format (**C**).

**Figure 7 viruses-16-00051-f007:**
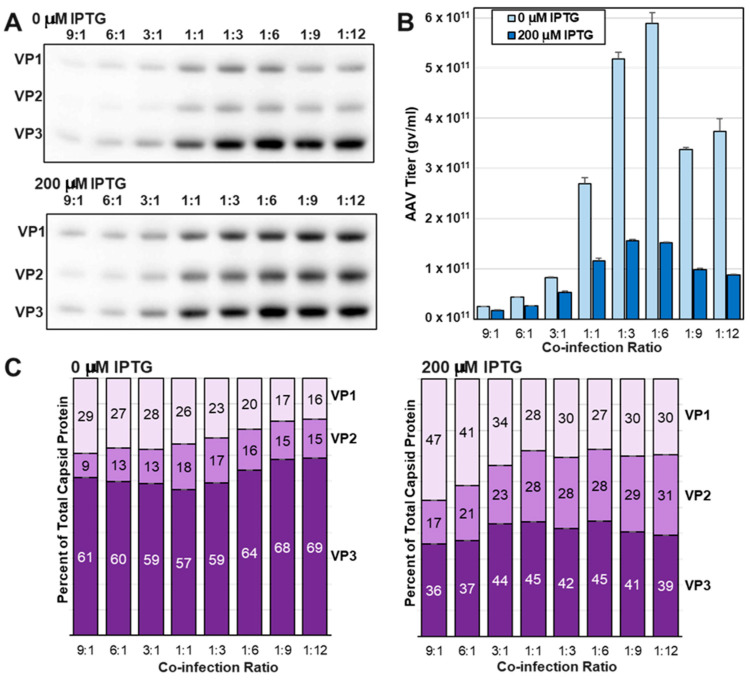
Optimizing VB-ITR-SEAP and VB-LacRepCap Bacmid Co-Infection Ratios. *Sf9* cells were co-infected with VB-ITR-SEAP and VB-LacRepCap bacmids at different co-infection ratios at 0.1 MOI in the presence of 0 μM and 200 μM IPTG. A 72 h post-infection, (**A**) Cell lysate protein samples were fractionated by SDS-PAGE and Western blot probed with anti-capsid antibody. (**B**) Crude cell lysate titers for ITR-SEAP-ITR transgenes were determined by Q-PCR. (**C**) Corresponding estimated capsid ratios are expressed in bar graph format for 0 μM IPTG and 200 μM IPTG.

**Figure 8 viruses-16-00051-f008:**
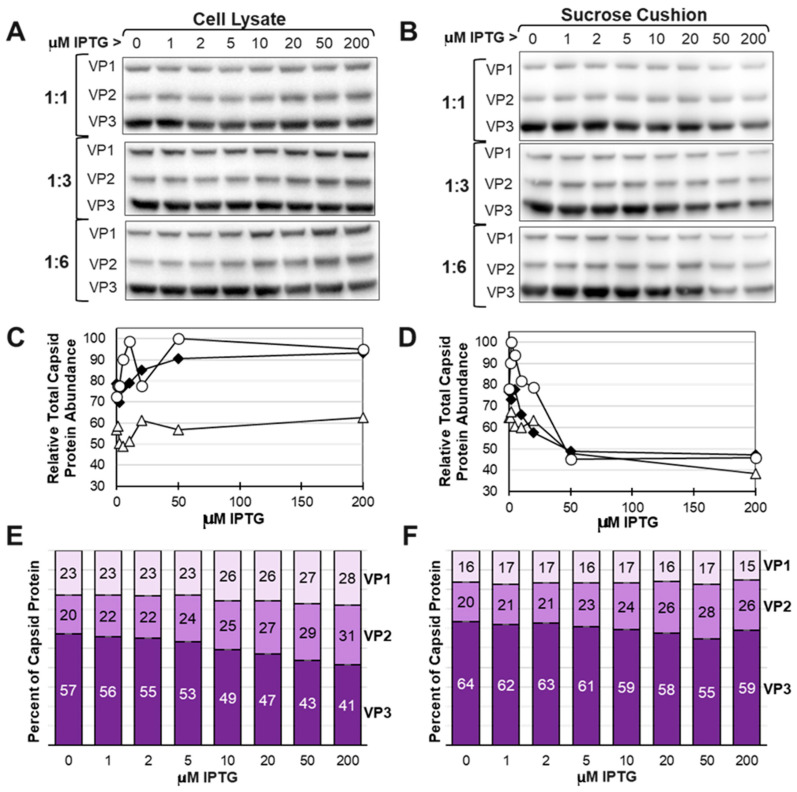
Effect of IPTG on total AAV Capsid Protein Abundance in Cells and in Purified AAV Capsids. *Sf9* cells were co-infected with VB-ITR-SEAP and VB-LacRepCap bacmids at 0.1 MOI and co-infection ratios of 1:1, 1:3, and 1:6 in the presence of different concentrations of IPTG. At 72 h post-infection, protein samples were obtained from cell lysates (**A**) and from sucrose cushion ultracentrifugation pellet fractions of cell lysates (**B**). All samples were fractionated by SDS-PAGE, and Western blots were probed with an anti-capsid monoclonal antibody. Immunolabeled capsid proteins were detected by ECL and quantified using ImageJ software. Total capsid protein abundances in cell lysates (**C**) and sucrose cushion pellets (**D**) were plotted relative to IPTG concentration for each co-infection group. Total capsid abundances at different co-infection ratios are represented on graphs as open triangles (1:1 ratio), black diamonds (1:3 ratio), and open circles (1:6 ratio). The estimated capsid ratios for the 1:6 co-infection are expressed in bar graph format for cell lysate (**E**) and sucrose cushion samples (**F**).

**Figure 9 viruses-16-00051-f009:**
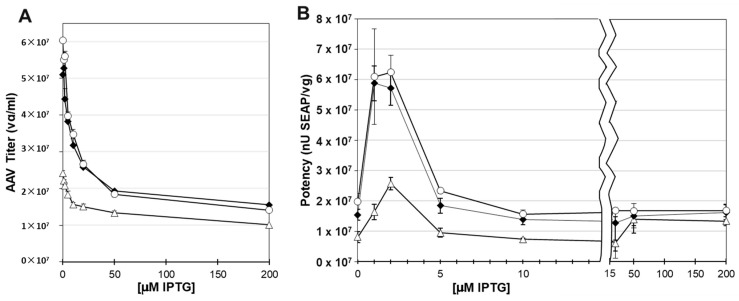
Titer and Potency of AAV Produced at Different IPTG Concentrations. *Sf9* cells were co-infected with VB-ITR-SEAP and VB-LacRepCap bacmids at 0.1 MOI. At 72 h post-infection, lysates were made. Sucrose cushion-purified AAV sample Q-PCR titers are plotted against IPTG concentration (**A**). Titers generated at ITR-SEAP-ITR:LacRepCap co-infection ratios of 1:1 (open triangle), 1:3 (black diamond), and 1:6 (open circle) are shown. Cells were lysed, and AAV capsids were purified by sucrose cushion ultracentrifugation. These AAV samples were used to transduce HEK 293T cells, and the alkaline phosphatase activity was measured after 4 days. Potency was measured as alkaline phosphatase activity relative to genome titer, which is plotted against IPTG concentration (**B**). The graph is split-scaled to emphasize the potency values of capsids produced at IPTG concentrations between 0 μM and 10 μM.

**Figure 10 viruses-16-00051-f010:**
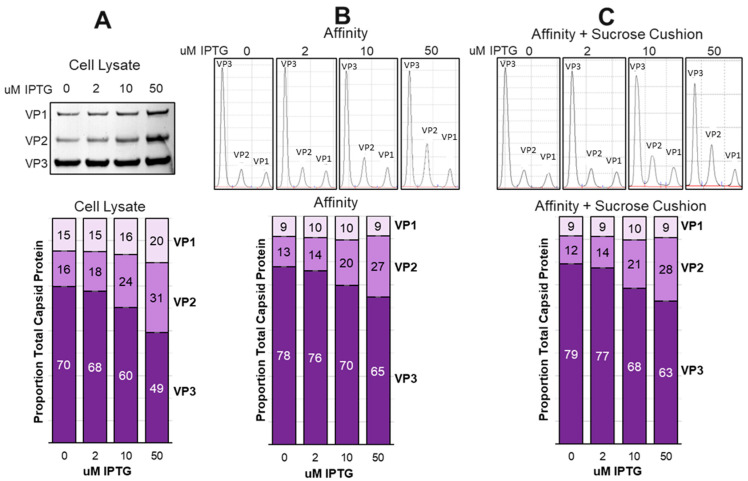
Comparison of Capsid Ratios Before and After Affinity Purification. *Sf9* cells were cultured at an 800 mL scale and co-infected at 0.001 MOI with VB-ITR-SEAP and VB-LacRepCap bacmids at a 1:6 co-infection ratio in the presence of 0 μM, 2 μM, 10 μM, and 50 μM IPTG. At 72 h post-infection, cells were lysed and capsids were affinity purified, followed by further sucrose cushion purification. Total capsid proteins in cell lysates were analyzed by Western blot and quantified from ECL images using ImageJ software (**A**). Affinity purified capsids (**B**) and affinity/sucrose cushion purified capsids (**C**) were analyzed by CE-SDS.

**Figure 11 viruses-16-00051-f011:**
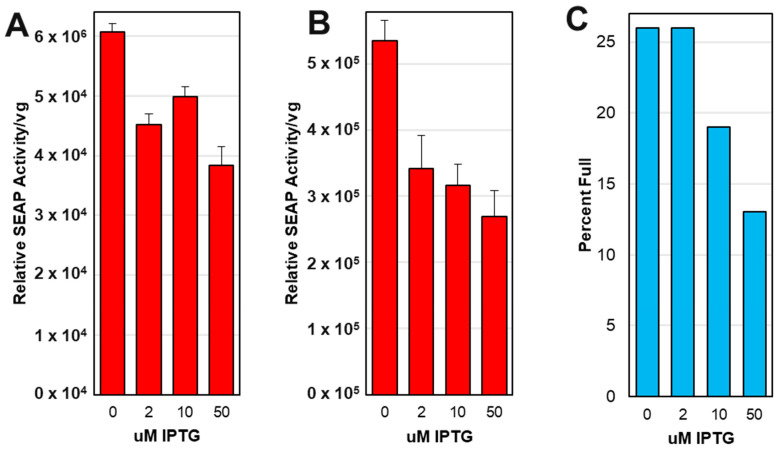
Potency and Percent Full of Affinity purified and affinity/sucrose cushion purified AAV capsids. Purified rAAV from VB-ITR-SEAP and VB-LacRepCap bacmid co-infection of *Sf9* cells at an 800 mL scale was used to transduce HEK 293T cells. The alkaline phosphatase activity was measured after 4 days. Potency was measured as alkaline phosphatase activity relative to AAV viral genome (vg) added to HEK 293T cells and is plotted against IPTG concentration for affinity purified AAV (**A**) and for affinity/sucrose cushion purified AAV (**B**). Affinity purified capsids were also subjected to SEC-MALS analysis, and the percent full (ITR-SEAP-ITR transgene genomes) in capsids relative to IPTG concentration is shown (**C**).

## Data Availability

Additional data require a confidential disclosure agreement (CDA) with Voyager Therapeutics.

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
