# Peer review of "A Lac Repressor-Inducible Baculovirus Expression Vector for Controlling Adeno-Associated Virus Capsid Ratios"

_viruses, 2023, doi:10.3390/v16010051_

Round 1

Reviewer 1 Report

Comments and Suggestions for Authors

This manuscript by Slack et al. demonstrated the production of rAAV using a baculovirus expression system that the ratio of expression AAV capsid proteins VP1,2,3 can be controlled by IPTG induction. The authors aimed to enhance the potency of rAAV produced via the BEV production system. Instead of expressing VP1-3 from a single cassette with an overlapping ORF (as in the conventional RepCap-BEV), they separately expressed the VP1 and 2 using the LacR-inducible promoters, while the VP3 was expressed via the non-inducible P10 promoter. This study could be of considerable interest to the field of rAAV production, as it highlighted how an overexpression of VP1 and VP2 expression in the production system adversely impacts the capsid assembly, likely attribute to the low percentage of the fully-packaged vector and subsequently reducing potency. However, a notable criticism is the lack of comparative analyses between the vectors produced from authors' optimized condition and those generated from the conventional BEV vector RepCap-BEV,  in both the vector potency and the ratio of the empty capsid/fully-packaged virion. Including such comparison in the revision would be appreciated.

Author Response

Thank you for the generally positive review.

We’ve chosen to not compare RepCap-BEV to the Lac inducible BEV as the purpose of the paper was to demonstrate an inducible design and not compare it directly with the original RepCap-BEV design.

Another reviewer requested extensive changes to make the manuscript more comprehensive. The revised version of the manuscript reports materials and methods and results in the order that constructs were generated.  Instead of describing constructs in one concise figure, they are now described chronologically.

Reviewer 2 Report

Comments and Suggestions for Authors

In this study, the authors described a method for expressing rAAV capsid VPs at different ratios for rAAV9 production in the BEV system. In this new system, VP1 and VP2 ORFs were driven under lac repressor (LacR) inducible regulation in the BEV expression system. The results showed the control of the abundances of VP1 and VP2, related to the VP3 level and how this ratio regulates the yield and potency of rAAV9. Overall, the abundant expression of VP1 and VP2 in fact negatively impacted capsids assembly and reduced the vector potency. Although the study did not identify an approach to get rAAV9 vector production or potency, it clarified some of the issues in BEV-based rAAV production and gained better understanding of the semi-stochastic nature of the assembly of AAV capsid.  

Author Response

Thank you for the positive response.

Another reviewer requested extensive changes to make the manuscript more comprehensive. The revised version of the manuscript reports materials and methods and results in the order that constructs were generated.  Instead of describing constructs in one concise figure, they are now described chronologically.

Comparison with the original RepCap baculovirus has also been removed as the main point of the manuscript is to demonstrate an inducible system.

Reviewer 3 Report

Comments and Suggestions for Authors

The manuscript describes production of adeno-associated virus (AAV) using the baculovirus expression vector (BEV) system. Since capsid proteins (VP1-3) of AAV need a certain stoichiometric ratio for proper assembly, high titer and potency, the amount of individual capsid proteins needs to be adjusted accordingly. The E. coli Lac repressor inducible system was applied for regulating the expression of VP1 & VP2 to change the properties of AAV. Overexpressed VP1 was disadvantageous and did not become incorporated, excess of VP2 reduced virus titer.

Production of AVV in baculovirus vectors (BV) is commonly based on the expression of capsid proteins from one ORF with alternative start codons to ensure a correct ratio for assembly. In this study an alternative approach is introduced which makes use of the E. coli lac repressor (LacR), part of the Lac Operon for controlling expression. Capsid proteins are cloned separately in the BV genome and VP1 & VP2 are placed under the strong BV p10 promoter flanked by two Lac Operon elements to allow for inducible regulation when a Lac repressor is expressed from the same BV. This idea is new in the context of BV and the functionality is shown in principle. There are many data in this study which I think should be more streamlined to enhance the presentation. Although it has merits I have several concerns.

1.      Fig1: all constructs used in the study are shown in material and methods section. Results start with a new construct (Fig 2), not described beforehand. Although it is depicted here and used to generally demonstrate if and to what extent LacR regulated expression works, it’s presented as intermediate construct (“see Methods and Fig 2A”). Apparently, the effect of repression was considered not being strong enough, since a second copy of the LacZ gene was included in the later versions of vectors. The description and arrangement of vector design is not consistent and confusing. This needs to be changed in a more logic flow.

2.      figure legends are of general concern. They only recapitulate the text description of material and methods, but do not explain the experiment, nor do they allow an interpretation.

3.      many parameters are tested in the experiments; e. g. capsid ratios which are depicted in almost every figure, partly lack sufficient explanation and contextual information. The influence of IPTG concentration, transgene/vector ratios, type of purification and the scale of production are compared and the results sometimes are conflicting.

4.      used terminology in the text and legend is not consistent: e.g. LacRepCap BEV = LacRepCap RepLacCap BEV= LacRepCap baculoviruses; ITR-SEAP-ITR=ITR-SEAP-BEV=ITR-SEAP, etc.). Needs to be standardized.

5.      what is the rationale of Fig 3? A time course (0-50 hpi) for LacRepCap BEV without IPTG is shown. Other experiments were done with samples harvested at 72 hpi. What is the relevance of VP1 expression peak at 14 hpi without IPTG? I don’t see the point?

Several figure legends lack experimental conditions of BV infection: MOI, time point of harvest.

6.      Fig 5: what is the message here? Unexpected observations are described. There is much speculation on multiple action of transcription factors and viral replication kinetics without showing evidence. The paragraph should be limited to the essential information.

7.      In Fig 6 a rapid purification method for AVV is introduced and compared to unpurified virus samples from cell lysate. This single step centrifugation substantially changes (reverses) capsid expression profile. What is the conclusion and how does that relate to the data from preceding experiments, since they are all based on cell lysates only?

The CMV promoter has been demonstrated to be active in Sf9 and other insect cells. Low background from the transgene signal is therefore not surprising.

8.      Fig 7: what is the time point when samples were harvested and subjected to sucrose cushion centrifugation? Include in Fig.

9.      Fig 8: Cell lysates in this experiment were affinity purified and a portion of those capsids were further purified by sucrose cushion ultracentrifugation. The authors mention only 1% AAV from cell lysate were recovered from sucrose pellet. What is the overall yield in case of affinity purification followed by sucrose cushion ultracentrifugation? I guess the yield is quite low. Was that the reason for using 800 mL culture volume in order to obtain enough material?

A head to head comparison of LacRepCap BEV and standard RepCap BEV, treated and purified under the same experimental conditions would be informative to evaluate a potential benefit of the LacO regulated construct.

10.  Fig 9: The authors describe that at large scale they could not confirm the findings from experiments done at 25 mL scale where a maximum potency of AAV was detected at 2 µM IPTG. At 800 mL scale the highest potency was without IPTG. Is that a valid comparison when samples were not purified in a comparable fashion in (Fig 7 vs. Fig 9), is that the reason for missing reproducibility? As demonstrated before, purification method has a huge impact on capsid protein abundance.

In summary, the authors demonstrate inducible gene regulation in baculovirus using the E. coli Lac repressor aimed at manipulating AAV capsid ratios of VP1 and VP2. General functionality of the system is shown, although for only a narrow range. Throughout the manuscript inconsistent capsid ratios are shown but I cannot recognize how LacR regulation contributes to desired properties (titer, potency). Method of purification and co-infection ratio do have significantly more influence. Many aspects are addressed thus what is the actual focus. If it is production yield, how does it compete with standard AAV expression, RepCap BEV? The conception of the manuscript seems premature, reminiscent of a step-by-step approach.

Again, I suggest to shorten and considerably condense the manuscript with a focus on the underlying objective.

minor:

line 178: change “based” into “based on”

line 185: 2.4. Sucrose Cushion rAAV Purification: What was the time point of harvest?

line 203: 2.5. Affinity Purification, CE-SDS Analysis, and Empty/Full Ratios: How many hpi was the supernatant harvested?

lines 210, 211: “At all steps in purification contained 0.001% v/v pluronic F-68 (#24040032, Thermo Fisher Scientific).“ Incomplete phrase.

line 219: 2.6. Western Blots.  How many hpi?

line 292: Fig 2: when was IPTG added?

line 307: change “the range repression” into “the range of repression”

line 342: change RepLacCap BEV“ into ” LacRepCap BEV“ Consistent naming.

line 347: change “cells ranged of ranged from” into “cells ranged from

line 356: Fig 2: include MOI into Fig legend.

line 396: “ITR-SEAP-GFP-ITR”  GFP?

line 435: “samples were obtained from cell lysates“ how many hours pi`?

lines 482 & 483: construct names “ITR-SEAP-GFP“ must be corrected. 2x

line 487: change “alkaline phosphatase activity relative to genome titer is and is plotted“ into “alkaline phosphatase activity relative to genome titer is plotted

line 510: name in Fig legend “ITR-SEAP-GFP“ must be corrected

lines 525 & 528: names in Fig legend “ITR-SEAP-GFP“ must be corrected

line 537: change “overlapping ORFs due the restricted“ into “overlapping ORFs; due to the restricted

line 538: change “genome capacity AAV” into “genome capacity of AAV”

line 547: change “into another unique lociinto “into another unique locus”:

line 564: change “percent full resulting” into “percent full capsids resulting

lines 568, 569; change “the impact of AAP expression as VP1 and VP2 expression was modulated by IPTG induction of VP1 and VP2 expression“ into “the impact of AAP expression as VP1 and VP2 expression was modulated by IPTG induction

line 574: change “result of leaking translational leaky scanning.“ into “result ot translational leaky scanning.

line 582: change “VP1 that which can be“ into “VP1 that can be

Author Response

  1. Fig1: all constructs used in the study are shown in material and methods section. Results start with a new construct (Fig 2), not described beforehand. Although it is depicted here and used to generally demonstrate if and to what extent LacR regulated expression works, it’s presented as intermediate construct (“see Methods and Fig 2A”). Apparently, the effect of repression was considered not being strong enough, since a second copy of the LacZ gene was included in the later versions of vectors. The description and arrangement of vector design is not consistent and confusing. This needs to be changed in a more logic flow.

I have revised the figures so that they follow a logical flow.

Figure 1. shows a schematic of the historical RepCap configuration in rBEV or Bacmids

Figure 2. shows original pMON14272 bacmid and the VoyBac1.1 bacmid used in this study

Figure 3. shows bacmid VB-LacR-LacOVP1 and its evaluation

Figure 4. shows bacmid VB-LacR-Rep-VP3, its modified promoter design and its evaluation

Figure 5. shows Bacmid VB-LacRepCap

Materials and Methods

2.3.2 Describes construction of bacmid VoyBac1.1

2.3.3 Describes construction of bacmid VB-LacR-LacOVP1

2.3.4 Describes construction of bacmid VB-LacR-Rep-VP3

2.3.5 Describes construction of bacmid VB-LacRepCap

Results

3.1 Describes results of experiments with VB-LacR-LacOVP1

3.2 Describes results of experiments with VB-LacR-Rep-VP3

3.3 and following sections describe experiments with VB-LacRepCap

  1. figure legends are of general concern. They only recapitulate the text description of material and methods, but do not explain the experiment, nor do they allow an interpretation.

Figure legends describe what is in the figures. Result and Discussion describe experiments and results and refer to the figures.

However I have modified titles in some legends to be more descriptive of experimental purpose

  1. many parameters are tested in the experiments; e. g. capsid ratios which are depicted in almost every figure, partly lack sufficient explanation and contextual information. The influence of IPTG concentration, transgene/vector ratios, type of purification and the scale of production are compared and the results sometimes are conflicting.

One of the main points of the paper is that scale of production affects the outcome.

I could have easily left off the larger scale data and it would have made a more perfect story.

However the audience involved in scale production of rAAV therapeutics it is needed information.

Not all trends were conflicting, and I made a point of describing this in the discussion.

For example saturation of VP1 in capsids happened both at 30ml scale Figure 85 and at 800 ml scale Figure 10B  and 10C.

  1. used terminology in the text and legend is not consistent: e.g. LacRepCap BEV = LacRepCap RepLacCap BEV= LacRepCap baculoviruses; ITR-SEAP-ITR=ITR-SEAP-BEV=ITR-SEAP, etc.). Needs to be standardized.

All has been standardized to Bacmid and the BEV references have been removed.

I also use the name acronym VB (VoyBac1.1) in front of all bacmid names.

VP-LacRepCap, VB-LacR-LacOVP1, VB-LacR-Rep-VP3, VB-LacRepCap

  1. what is the rationale of Fig 3? A time course (0-50 hpi) for LacRepCap BEV without IPTG is shown. Other experiments were done with samples harvested at 72 hpi. What is the relevance of VP1 expression peak at 14 hpi without IPTG? I don’t see the point?

I’ve clarified the rational for this experiment by adding explanation to results section 3.2

  “Baculovirus gene expression has a distinct temporal cascade of early (6h), late (12h) and hyper expressed very late gene expression (24h). We decided to express LacR from an earlier baculovirus promoter which would not compete with the very late baculovirus promoters driving AAV capsid VP expression.”

Several figure legends lack experimental conditions of BV infection: MOI, time point of harvest.

 I’ve included MOI’s and infection times

  1. Fig 5: what is the message here? Unexpected observations are described. There is much speculation on multiple action of transcription factors and viral replication kinetics without showing evidence. The paragraph should be limited to the essential information.

Yes that is why we clearly say “speculate” in the results section. The key point is the location in the bacmid appears to be affecting expression and complicating the system and this is important to note to provide a possible explanation relating to the well characterized transcriptome around the regions of cloning.

I’ve included an additional observation.

“When we alternated the locations of LacOVP1 and LacOVP2, there was more VP2 expression that VP1 (data not shown). “

  1. In Fig 6 a rapid purification method for AVV is introduced and compared to unpurified virus samples from cell lysate. This single step centrifugation substantially changes (reverses) capsid expression profile. What is the conclusion and how does that relate to the data from preceding experiments, since they are all based on cell lysates only?

No that is not what the data show. Lysates are total capsid proteins and sucrose cushions are assembled capsids. Lysates will contain capsid protein monomers and capsids. Not all capsid proteins expressed get assembled into capsids. Saturating expression with VP1 results in less capsid assembly and thus lower total capsid yields.

Further more the figure shows there limit to the amount of VP1 which can be incorporated into capsids.

The CMV promoter has been demonstrated to be active in Sf9 and other insect cells. Low background from the transgene signal is therefore not surprising.

Retention of SEAP enzyme on baculovirus capsids passing through sucrose cushion was surprising. I deleted the part mentioning CMV promoter

“We discovered that there was SEAP expression by the VB-ITR-SEAP bacmid in Sf9 cells and resulting SEAP enzyme may be associated with baculovirus capsids co-purifying with AAV capsids through sucrose cushions.”

  1. Fig 7: what is the time point when samples were harvested and subjected to sucrose cushion centrifugation? Include in Fig.

72h post infection harvest, 0.1 MOI infection

  1. Fig 8: Cell lysates in this experiment were affinity purified and a portion of those capsids were further purified by sucrose cushion ultracentrifugation. The authors mention only 1% AAV from cell lysate were recovered from sucrose pellet. What is the overall yield in case of affinity purification followed by sucrose cushion ultracentrifugation? I guess the yield is quite low. Was that the reason for using 800 mL culture volume in order to obtain enough material?

The yield from affinity purification is substantially higher but time and effort to do this larger scale production is much greater. The point of the sucrose purification is to enable a large number of conditions to be evaluated. Unfortunately, it did not scale up. That is key point in this paper. As I’ve said ealier the scaled up data could have easily been omitted to make a deceptive result.

A head to head comparison of LacRepCap BEV and standard RepCap BEV, treated and purified under the same experimental conditions would be informative to evaluate a potential benefit of the LacO regulated construct.

This comparison has been removed because the point of the paper is to show what happens to rAAV when VP1 and VP2 are regulated by Lac Repressor at small scale and at large scale.

  1. Fig 9: The authors describe that at large scale they could not confirm the findings from experiments done at 25 mL scale where a maximum potency of AAV was detected at 2 µM IPTG. At 800 mL scale the highest potency was without IPTG. Is that a valid comparison when samples were not purified in a comparable fashion in (Fig 7 vs. Fig 9), is that the reason for missing reproducibility? As demonstrated before, purification method has a huge impact on capsid protein abundance.

I’ve altered the discussion.

IPTG induction of LacR repression of VP1 and VP2 expression relative to VP3 was tunable such that the potency of resulting capsids could be optimized in small 25 mL scale Sf9 cultures. This did not translate to larger 800 mL scale production with the non-induced LacR repression producing the most potent rAAV. A possible reason for this was that the VB-ITR-SEAP bacmid and VB-RepLacCap bacmid co-infection ratios at larger scale needed to be further optimized. As shown in Figure 9B, having too little VB-LacRepCap bacmid relative to VB-ITR-SEAP bacmid reduced the ability to tune potency with IPTG. The 800 ml scale VB-ITR-SEAP and VB-LacRepCap bacmid co-infections were also carried out an MOI about 100-fold lower than the 25 ml scale co-infections. This lower MOI may have generated larger populations of infected Sf9 cells with suboptimal co-infection ratios based on the normal distribution of two co-infecting bacmids [43].   

[43]Pazmiño-Ibarra V, Herrero S, Sanjuan R. Spatially Segregated Transmission of Co-Occluded Baculoviruses Limits Virus-Virus Interactions Mediated by Cellular Coinfection during Primary Infection. Viruses. 2022 Jul 31;14(8):1697.

In summary, the authors demonstrate inducible gene regulation in baculovirus using the E. coli Lac repressor aimed at manipulating AAV capsid ratios of VP1 and VP2. General functionality of the system is shown, although for only a narrow range. Throughout the manuscript inconsistent capsid ratios are shown but I cannot recognize how LacR regulation contributes to desired properties (titer, potency). Method of purification and co-infection ratio do have significantly more influence. Many aspects are addressed thus what is the actual focus. If it is production yield, how does it compete with standard AAV expression, RepCap BEV? The conception of the manuscript seems premature, reminiscent of a step-by-step approach.

Again, I suggest to shorten and considerably condense the manuscript with a focus on the underlying objective.

Accommodation of this review made the manuscript longer.

minor:

line 178: change “based” into “based on”

line 185: 2.4. Sucrose Cushion rAAV Purification: What was the time point of harvest?

(in figure 8)

 At 72h post infection, protein samples were obtained from cell lysates (A) and from sucrose cushion ultracentrifugation pellet fractions of cell lysates

line 203: 2.5. Affinity Purification, CE-SDS Analysis, and Empty/Full Ratios: How many hpi was the supernatant harvested?

(in figure 10)

At 72h post infection cells were lysed and capsids were affinity purified and then further sucrose cushion purified. 

lines 210, 211: “At all steps in purification contained 0.001% v/v pluronic F-68 (#24040032, Thermo Fisher Scientific).“ Incomplete phrase.

“All steps in purification contained 0.001% v/v pluronic F-68 (#24040032, Thermo Fisher Scientific).“

line 219: 2.6. Western Blots.  How many hpi?

hpi is described in figures of Westerns.

line 292: Fig 2: when was IPTG added?

Sf9 cells were infected at 10 MOI in presence of different concentrations of IPTG. At 72 hpi, infected Sf9 cells were collected

line 307: change “the range repression” into “the range of repression”

line 342: change “RepLacCap BEV“ into ” LacRepCap BEV“ Consistent naming.

Naming has be standardized  ie VP-LacRepCap bacmid

line 347: change “cells ranged of ranged from” into “cells ranged from“

ratios in infected Sf9 cells ranged from 15:17:68 at 0 uM IPTG to 30:32:38 at 50 uM and 100 uM IPTG 

line 356: Fig 2: include MOI into Fig legend.

 Sf9 cells were infected at 10 MOI

line 396: “ITR-SEAP-GFP-ITR”  GFP?

The GFP was a mistake and GFP is removed

line 435: “samples were obtained from cell lysates“ how many hours pi`?

At 72h post infection, protein samples were obtained from cell lysates

lines 482 & 483: construct names “ITR-SEAP-GFP“ must be corrected. 2x

The GFP was a mistake and GFP is removed

line 487: change “alkaline phosphatase activity relative to genome titer is and is plotted“ into “alkaline phosphatase activity relative to genome titer is plotted“

 alkaline phosphatase activity relative to genome titer is plotted

line 510: name in Fig legend “ITR-SEAP-GFP“ must be corrected

The GFP was a mistake and GFP is removed

lines 525 & 528: names in Fig legend “ITR-SEAP-GFP“ must be corrected

The GFP was a mistake and GFP is removed

line 537: change “overlapping ORFs due the restricted“ into “overlapping ORFs; due to the restricted“

In nature, VP1, VP2, and VP3 evolved to be overlapping ORFs due to the restricted genome capacity of AAV.

line 538: change “genome capacity AAV” into “genome capacity of AAV”

In nature, VP1, VP2, and VP3 evolved to be overlapping ORFs due to the restricted genome capacity of AAV.

line 547: change “into another unique loci“ into “into another unique locus”:

We also cloned and expressed the E. coli LacR gene into another unique locus in the bacmid.

line 564: change “percent full resulting” into “percent full capsids resulting“

show a decline in the percent full capsids resulting from increased expression 

lines 568, 569; change “the impact of AAP expression as VP1 and VP2 expression was modulated by IPTG induction of VP1 and VP2 expression“ into “the impact of AAP expression as VP1 and VP2 expression was modulated by IPTG induction “

Both VP1 and VP2 ORFs have complete copies of the AAP ORF. We did not address how modulating VP1 and VP2 expression affected AAP expression.

line 574: change “result of leaking translational leaky scanning.“ into “result ot translational leaky scanning.“

However, as VP1 and VP2 transcription increased with IPTG induction, there would be more AAP produced because of translational leaky scanning. 

line 582: change “VP1 that which can be“ into “VP1 that can be“

an upper limit to the amount of VP1 that can be included

Reviewer 4 Report

Comments and Suggestions for Authors

Slack et al have designed and carried out an effective study to develop a more optimized rAAV production system with a dual BEV approach. The authors have done a tremendous job of both design and communication: their introduction is thorough, and the data presented thoroughly demonstrate the output of the work. The figures are clear with legible and relevant text, while figure captions are complete. The methods were moderately clear. While the system designed is not perfect, overall I find no issues with the study and only one minor issue with the manuscript. This paper represents work that should be useful in the optimization and eventual increased utility of dual BEVs for production of rAAVs. 

My only minor issue is in regards to details of the synthesis of constructs. For example, in section 2.3.2 the authors state that the LacR cassettes were “made” with no detail here; in section 2.3.4 they state that “cassette(s) for T4 ligation was cut from synthetically made donor plasmids…” without stating nature of the donor plasmids; in 2.3.6 they state that the donor plasmid was synthesized by GenScript. Were the other constructs similarly synthesized?

Author Response

Thank you for the positive response.

I’ve made the minor changes requested.

Another reviewer requested extensive changes to make the manuscript more comprehensive. The revised version of the manuscript reports materials and methods and results in the order that constructs were generated.  Instead of describing constructs in one concise figure, they are now described chronologically.

Comparison with the original RepCap baculovirus has also been removed as the main point of the manuscript is to demonstrate an inducible system.

Round 2

Reviewer 1 Report

Comments and Suggestions for Authors

Reviewer agrees with the publication of this vision.